# Zero-Shot Coordination via Semantic Relationships Between Actions and Observations

## Abstract

An unaddressed challenge in zero-shot coordination (ZSC) regards taking advantage of the semantic relationship between the features of an action and the features of observations. Humans take advantage of these relationships in highly intuitive ways. For instance, we might point to the object we desire or hold up fingers to indicate how many objects we want. This work investigates the effect of network architectures on the propensity of learning algorithms to make use of these relationships in human-compatible ways. We find that attention-based architectures that jointly process a featurized representation of the observation *and* the action have a better inductive bias for exploiting semantic relationships for zero-shot coordination. Excitingly, in a set of diagnostic tasks, these agents produce highly human-compatible policies, without requiring the symmetry relationships of the problems to be hard-coded.

## 1 Introduction

Successful collaboration between agents requires coordination (Tomasello et al., 2005; Misyak et al., 2014; Kleiman-Weiner et al., 2016), which is challenging because coordinated strategies can be arbitrary (Lewis, 1969; Young, 1993; Lerer & Peysakhovich, 2018). A priori, one can neither deduce which side of the road to drive, nor which utterance to use to refer to ♡ (Pal et al., 2020). In these cases coordination can arise from actors best responding to what others are already doing i.e., following a convention. For example, Americans drive on the right side of the road and say "heart" to refer to ♡ while Japanese drive on the left and say "shinzo". Yet in many situations prior conventions may not be available and agents may be faced with entirely novel situations or partners. In this work we study ways that agents may learn to leverage abstract relations between observations and actions to coordinate with agents they have had no experience interacting with before.

To illustrate, consider the following situations where people can figure out how to coordinate without prior experienced or shared conventions. Imagine a store that sells strawberries and blueberries. You want to buy strawberries but you don't share any common language with the clerk. You are however wearing a red hat and you wave the hat at the clerk to hint that the strawberries are what you want. The clerk has two baskets of strawberries remaining, and so you raise a single finger to indicate that you only want one of the baskets. The clerk produces a paper and plastic bag and you point to the paper bag to indicate that you want the paper one. These examples are so simple that they seem obvious: the red hat matches the colors of the strawberries, the number of fingers matches the number of baskets you want, and you extend a finger in the direction of the desired packaging (Grice, 1975). While obvious to people, who rely on a theory-of-mind in understanding others, we will show that these inferences remain a challenge for multi-agent reinforcement learning agents.

Less obvious examples are common in the cognitive science literature. Consider the shapes in Fig. 1. When asked to assign the names "Boubo" and "Kiki" to the two shapes people name the jagged object "Kiki" and the curvy object "Boubo" (Köhler, 1929). This finding is robust across different linguistic communities and cultures and is even found in young children (Maurer et al., 2006). The causal explanation is that people match a "jaggedness"-feature and "curvey"-feature in both the visual and auditory data. Across the above these cases, there seem to be a generalized mechanism for mapping the features of the persons action with the features of the desired action. All are examples of where in the absence of norms or conventions, people minimize the distance between features when making a choice. This basic form of *zero-shot coordination* predates verbal

behavior (Tomasello et al., 2007) and this capability has been hypothesized as a key predecessor to more sophisticated language development and acquisition (Tomasello et al., 2005). Modeling these capacities is key for building machines that can robustly coordinate with other agents and with people (Kleiman-Weiner et al., 2016; Dafoe et al., 2020).

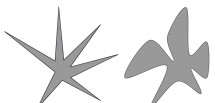

Figure 1: When asked to assign the names "Boubo" and "Kiki" to these two shapes, people systematically assign the name "Boubo" to the jagged object on the left and "Kiki" to the curvy object on the right.

However, as we will show, naively training reinforcement learning (RL) agents with self-play fails to learn to coordinate even in these obvious ways. Instead, they develop arbitrary private languages that are uninterpretable to both the same models trained with a different random seed as well as to human partners (Hu et al., 2020). For instance in the examples above, they would be equally likely to wave a red-hat to hint they want strawberries as they would to indicate that they want blueberries.

These problems also emerge at scale in the decentralized partially observable Markov decision process (Dec-POMDP) benchmark Hanabi (Bard et al., 2019). When agents are trained with self-play using standard architectures they do not develop strategies that take into account the correspondence between the features of the actions (colored and numbered cards) and the observation of the game state (other colored and numbered cards). Unfortunately, describing the kind of abstract knowledge that these agents lack in closed form is challenging. Rather than attempting to do so, we take a *learning-based* approach. Our aim is to build an agent with the capacity to develop these kinds of abstract correspondences during self-play such that they can robustly succeed during cross-play or during play with humans.

Our contributions are as follows: (1) We extend Dec-POMDPs to allow actions and observations to be represented using shared features and develop a novel human-interpretable environment for studying coordination in this setting. (2) We evaluate the role of neural network (NN) architectures including feedforward, recurrent, and attention mechanisms on both cross-play generalization and ability to create human-interpretable conventions. (3) We show that an attention architecture which takes *both* the action and observations as input allows the agent to exploit the semantic relationships for coordination, resulting in strong cross-play and human compatible policies that outperform baseline ZSC methods. This model also demonstrates sophisticated coordination patterns that exploit mutual exclusivity and implicature, two well-known phenomena studied in cognitive science (Markman & Wachtel, 1988; Grice, 1975).

## 2 RELATED WORK

**Cooperative MARL.** The standard method for training multi-agent reinforcement learning (MARL) agents in fully cooperative, partially observable settings is self-play (SP). However, the failure of SP policies in cross-play (XP) has been recently explored. Carroll et al. (2019) used grid-world MDPs to show that both SP and population-based training fail when paired with human collaborators. In Hanabi, agents trained via SP develop conventions that fail to generalize to independently trained agents from the same algorithm with different random seeds (Bard et al., 2019; Hu et al., 2020).

**Zero-Shot Coordination.** To address this issue, Hu et al. (2020) formally introduced the zero-shot coordination (ZSC) framework, where the goal is to maximize the XP returns of independently trained agents, allowing them to coordinated at test time. Thus formulated, ZSC is an alternative to ad-hoc teamplay, a framework for measuring coordinated team success when faced with players with unknown behavior (Stone et al., 2010; Barrett et al., 2011), which can be formalized as playing a best response to a distribution of a-priori known agents.

A few methods have attempted to address the ZSC framework. Other-Play (OP) (Hu et al., 2020) exploits the symmetries in a given Dec-POMDP to prevent agents from learning equivalent but mutually incompatible policies. OP prohibits arbitrary tie-breaking, thereby preventing equivalent conventions from forming. However, OP requires experimenter-coded symmetries, and discovering such symmetries is computationally challenging. In contrast, our learning based approach requires no experimenter-coding. Another recent method, Off-Belief Learning (OBL) (Hu et al., 2021), regularizes agents' ability to make inferences based on the behavior of others. Compared to prior

work on Hanabi where SP scores were high but XP scores were near chance, both of OP and OBL drastically improve XP scores and show promising preliminary results in play with people. However, neither of these algorithms can exploit the correspondences between features of an action and the observation of the state as we show in this work, unless this falls out of the environment dynamics.

**Attention for Modeling Input-Output Relationships.** Attention (Vaswani et al., 2017; Bahdanau et al., 2015; Xu et al., 2016) is an important tool for large sequence models, and exploiting semantic relationships between inputs and outputs via an attention-based model has been studied in the deep learning literature. In natural language processing, such an idea is commonly applied to question answering models (dos Santos et al., 2016; Tan et al., 2016; Yang et al., 2016). For instance, Yang et al. (2016) form a matrix that represents the semantic matching information of term pairs from a question and answer pair, and then use dot-product attention to model question term importance. For regression tasks, Kim et al. (2019) proposed Attentive Neural Processes (ANP) that use dot-product attention to allow each input location to attend to the relevant context points for the prediction, and applied ANP to vision problems. However, to our knowledge, we are the first to apply attention to exploit shared features of actions and observations in a Dec-POMDP setting for coordination.

**Human Coordination.** Our work is also inspired by how humans coordinate in cooperative settings. Theory-of-mind, the mechanism people use to infer intentions from the actions of others, plays a key role in structuring coordination (Wu et al., 2021; Shum et al., 2019). In particular, Rational Speech Acts (RSA) is a influential model of pragmatic implicature (Frank & Goodman, 2012; Goodman & Stuhlmüller, 2013). At the heart of these approaches are probabilistic representations of belief which allow for the modeling of uncertainty and recursive reasoning about each others beliefs, enabling higher-order mental state inferences. This recursive reasoning step also underlies the cognitive hierarchy and level-K reasoning models, and is useful for explaining certain focal points (Camerer, 2011; Stahl & Wilson, 1995; Camerer et al., 2004). However, constructing recursive models of players beliefs and behavior is computationally expensive as each agent must construct an exponentially growing number of models of each agent modeling each other agent. As a result, recursive models are often limited to one or two levels of recursion. Furthermore, none of these approaches can by itself take advantage of the shared features across actions and observations.

## 3 BACKGROUND

**Dec-POMDPs.** We use decentralized partially observable Markov decision processes (Dec-POMDPs) to formalize our setting (Nair et al., 2003). In a Dec-POMDP, each player $i$ observes the underlying state $s$ partially through an observation function $\Omega^i(s) \in \mathcal{O}^i$, and takes action $a^i \in \mathcal{A}^i$. Players receive a common reward $R(s, a)$ and the state follows the transition function $\mathcal{T}(s, a)$. The historical trajectory is denoted as $\tau = (s_1, a_1, \ldots, a_{t-1}, s_t)$. Player $i$'s *action-observation history* (AOH) is denoted as $(\Omega^i(s_1), a_1^i, \ldots, a_{t-1}^i, \Omega^i(s_t))$. The policy for player $i$ takes as input an AOH and outputs a distribution over actions, denoted by $\pi^i(a^i \mid \tau_t^i)$. The joint policy is denoted by $\pi$.

**Dot-Product Attention.** Given a sequence of input vectors $(x_1, ..., x_m)$, dot-product attention uses three weight matrices $(Q, K, V)$ to obtain triples $(Qx_i, Kx_i, Vx_i)$ for each $i \in \{1, \ldots, m\}$, called query vectors, key vectors, and value vectors. We abbreviate these as $(q_i, k_i, v_i)$. Next, for each $i, j$, dot-product attention computes logits using dot products $q_i \cdot k_j$. These logits are in turn used to compute an output matrix

$$[\text{softmax}(q_i \cdot k_1/\sqrt{m}, \ldots, q_i \cdot k_m/\sqrt{m}) \cdot v_j]_{i,j}.$$

We denote this output matrix as $Attention(x_1, \ldots, x_m)$.

## 4 DEC-POMDPS WITH SHARED ACTION AND OBSERVATION FEATURES

It is common to describe the states and observations in Dec-POMDPs using features, e.g. in games each card has a rank and a suit. These featurized observations can be readily exploited by function approximators. In contrast, in typical implementations the actions are outputs of the neural network and the models do not take advantage of features of the actions. In the standard representation of Dec-POMDPs, actions are defined solely through their effect on the environment. In contrast, in real world environments they are commonly grounded and can be described with semantic features referring to the object they are action on, e.g. "I pull the *red lever*".

To allow action features be used by the models, we first formalize the concept of observation and action features in Dec-POMDPs. We say a Dec-POMDP has *observation features* if for at least one player $i$, we can factor the observation $\Omega^i(s)$ into $l$ objects $\Omega^i(s) = (O_1, \ldots, O_l)$, where each object $O_j = \{f_1, \ldots, f_{n_j}\}$ is described by at most $n \geq n_j$ features. Each of these features $f_k$, where $k = 1, \ldots, n_j$, exists in a feature space $F_k$. Similarly, a Dec-POMDP has *action features* if one can factor the representation of the actions. In this case, at time $t$, the action player $i$ takes can be represented as $a_t^i = (\hat{f}_1, \ldots, \hat{f}_m)$, where each action feature $\hat{f}_r \in \hat{F}_r$, $r = 1, \ldots, m$, and $\hat{F}_r$ is the action feature space. Note that this representation typically is a function of the current state, e.g. an agent might always have 5 actions available (pulling a given lever), but the color of the levers available is contextual.

In some Dec-POMDPs the actions can further be described using some of the *same* features that describe the observations, for example when an agent can both observe the "red" light and pull the "red" lever. In these cases there is a *non-empty intersection* between $F_k$ and $\hat{F}_r$ ("shared action-observation features") which can be exploited for coordination. We note that in generality the action space may also include actions that act on sets of objects. While our formalism can be naturally extended to cover this, we will omit it here for brevity since it's not needed for our examples.

## 5 THE HINT-GUESS GAME

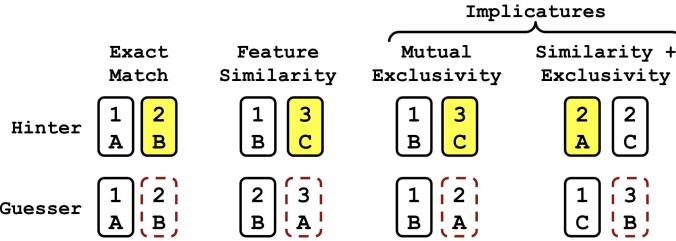

Figure 2: Example scenarios in the hint-guess game. The *Hinter* and *Guesser* are dealt a hand of cards where each card has a randomly sampled letter [A-C] and number [1-3]. All cards are visible to both players. Only the *Hinter* knows which of the *Guesser's* cards is the target card, outlined with a dotted red line above. The goal of the game is for the *Guesser* to select the target card but the only actions the *Hinter* can take is to point to one of their own cards. Shown above are four hand crafted scenarios that test distinct dimensions important for zero-shot coordination. The highlighted yellow card corresponds to a human interpretable choice. The two right scenarios require agents to reason about implicatures, i.e., the interpretable choice has zero feature overlap with the target card. Model performance is shown in Table 2.

We develop a novel Dec-POMDP with shared actions and observation features to study coordination in this setting. Our environment is two player game called the "hint-guess" game where players must coordinate in order to successfully guess a target card. The game consists of a *Hinter* and a *Guesser*. Both players are given a hand of $N$ cards, $H_1 = \{C_1^1, \ldots C_N^1\}$ for the *Hinter* and $H_2 = \{C_1^2, \ldots C_N^2\}$ for the *Guesser*. Each card is marked by two features $(f_1, f_2)$ where $f_1 \in F_1$ and $f_2 \in F_2$. Cards in each hand are drawn independently and randomly with replacement, with equal probability for any feature combination. Both hands, $H_1$ and $H_2$, are public information and are exposed to both players. Before the game starts, one card of the *Guesser*, $C_i^2$, is randomly chosen to be the target card and its features are revealed to the *Hinter*, but not the *Guesser*.

In the first round, the *Hinter* (who observes $H_1, H_2, C_i^2$) chooses a card of its own, call it $C_j^1$, to show to the *Guesser*. And in the second round, the *Guesser* (who observes $H_1, H_2$ and the features of $C_j^1$) guesses which of its cards is the target. Both players receive a common reward $r = 1$ if the features of the card played match those of the target, and otherwise $r = 0$.

Figure 2 shows some simple scenarios that probe key dimensions of coordination with $N = 2$, $F_1 = \{1, 2, 3\}$ and $F_2 = \{A, B, C\}$. Each of these scenarios have a human-interpretable solution. The first scenario (Exact Match) is the most simple, the *Hinter* has a copy of the target card, 2B, so they can just hint 2B. Note that while this strategy is human interpretable, the opposite convention would work in this particular case. The next scenario (Feature Similarity) requires reasoning about

the features under some ambiguity since none of the cards in the two hands are a direct match. In this case, both cards in the *Hinter's* hand share one feature with the *Guesser*. The human-interpretable strategy would be to match the cards that share features to each other. Although like before the exactly opposite strategy would also work well.

We also develop stimuli that require more sophisticated inferences (labeled implicatures in Figure 2). Implicatures require understanding the action embedded within its context. The third scenario (mutual exclusivity) invokes a simple kind of implicature: mutual exclusivity. In this scenario, human-interpretable reasoning follows the logic of: "if the target card *was* 1B, the *Hinter* would choose 1B so that means 1B is taken and 3C should correspond to 2A even though they share zero feature overlap. The final scenario combines feature similarity and mutual exclusivity. These scenarios are particularly interesting because deep learning models often struggle to effectively grapple with mutual exclusivity (Gandhi & Lake, 2020).

## 6    THE EFFECT OF ARCHITECTURE CHOICE ON ZERO-SHOT COORDINATION

We consider the following architectures to investigate the effect of policy parameterization on the agents' ability to exploit shared action-and observation features for ZSC.

**Deep Q-Learning (DQN).** We first use standard DQN with feedforward neural networks (NN). All observations are concatenated and fed into an NN, which outputs the estimated Q-value for each action. There is no explicit representation of action-observation relationships in this model, since observations are inputs and actions are outputs. E.g. for multi-layer perceptrons (MLP) the inputs and outputs can be permuted arbitrarily whiteout changing the learning problem.

**Recurrent Deep Q-Learning (RDQN).** In the RDQN model, we feed in objects in the observation (namely, vectors representing cards) sequentially to a long short-term memory (LSTM) network (Hochreiter & Schmidhuber, 1997). To improve trainability we concatenate all hidden states from each step and use that as input to a feedforward NN, noting that this is unconventional. Like the MLP, the LSTM does not explicitly model the relationship between action and observation features.

**Attention Mechanisms.** The core of attention is the dot product, which is fundamentally a comparative operator. As such, it is natural to explore attention for ZSC in our setting and in the following we present three different options for attention-based architectures. While the attention mechanism is the same for all three architectures, they differ in terms of their overall design and we will see that these differences are highly relevant for the learning outcomes. Let $Attention(\cdot)$ denote the attention layer as introduced in Section 3.

**Attention for Observation (Obs).** The first model feeds observations into the attention layer, then the flattened output is passed through a simple ReLU feedforward layer and produces the estimated Q-values for all actions as an output.

$$S = Attention(O_1, \ldots, O_n)$$
$$P = ReLU(S),$$

where $P$ are the estimated Q-values for all of the actions.

**Attention with Action as Output (Action-Out).** In order to more explicitly represent the relational structure, we take the dot product of a linear transform of the attention layer and an embedding vector for each of the actions.

$$S = Attention(O_1, \ldots, O_n)$$
$$P = Linear(S) \cdot Linear(A),$$

where $Linear(X) = XW + b$, where $W$ is the weight matrix and $b$ is the bias term.

**Attention with Action as Input (Action-In).** Next we extend this architecture to take advantage of shared features between observations and actions. We feed a feature based representation for a given action (one at a time) into the attention mechanism along with the observation. This outputs a single scalar value at a time, the estimated Q-value for the specific action being fed into the network.

$$S_k = Attention(O_1, \ldots, O_n, A_k) \quad \text{for } k = 1, \ldots, m$$
$$P = [Linear(S_1), \ldots, Linear(S_m)].$$

To be clear, this architecture will take $N$ forward passes (one per action) to calculate the Q-values for all actions.

## 7 EXPERIMENTS

We evaluate the architectures in the hint-guess game mentioned in Section 5. We fix the hand size $N = 5$ and the features to be $F_1 = \{1, 2, 3\}$ and $F_2 = \{A, B, C\}$. As shown in Appendix A.4, similar results can be obtained with a larger or smaller hand size. We use a two-hot vector to represent the features of each card. The observation input is a concatenation of these representations for both hands, $H_1$ and $H_2$, with the vector representing the target card, $C_i^2$ and hinted card $C_j^1$. We evaluate the agents' performance and behavior in both self-play, zero-shot coordination setting, and ability to match the human interpretable response in four scenarios.

We train agents in the standard self-play setting using independent Q-learning (Tan, 1993), where the *Hinter* and *Guesser* are jointly trained to maximize their score in 4M randomly initialized games. To avoid giving the set-based attention architectures an unfair advantage, we independently permute the cards observed by the agents so they are not able to coordinate using the position of the cards. More details about training setup and the model architectures can be found in Appendix A.1.

## 8 RESULTS

### 8.1 CROSS-PLAY PERFORMANCE.

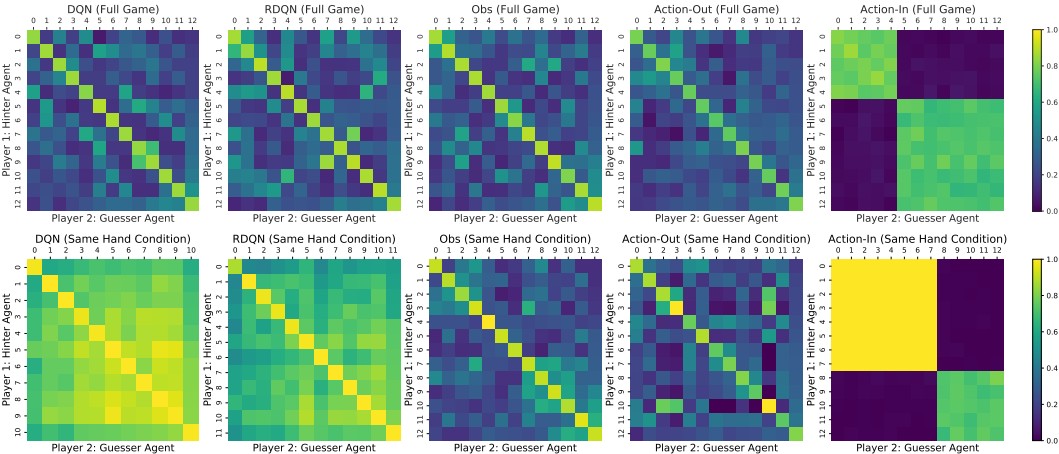

Figure 3: Cross-play performance. Visualization of paired evaluation of different agents trained under the same method. The top row is the result for the full hint-guess game and the bottom row is the result for a simplified game where the *Hinter* and *Guesser* agents are always dealt the exact same hand. The y-axis represents the agent index of the *Hinter* and the x-axis represents the agent index of the *Guesser*. Each block in the grid is obtained by evaluating the pair of agents on 10K games with different random seeds. Numerical performance is shown in Table 1.

First, we evaluate model performance for each architecture in the cross-play (XP) setting. In this setting, agents from independent training runs with different random seeds are paired together. Clearly, high XP performance is a prerequisite or necessary (but not sufficient) condition for coordinating with other AI agents or humans. If agents trained from independent runs using the same model and training algorithm cannot coordinate with each other, it is highly unlikely they will be able to coordinate with agents built with different architectures or trained using different procedures and even less likely they will be able to coordinate with people (Hu et al., 2020).

Figure 3 records the scores obtained by each pair of agents, where the diagonal entries are the within-pair SP scores and the off-diagonal entries are XP scores. The bottom row shows the same results for a simplified version of the game, where both *Hinter* and *Guesser* have the same hand. In the

simplified game, an optimal and intuitive strategy exists, which is to always hint the target card and then guess the card that is hinted. Table 1 summarizes average SP and XP scores across agents.

From the top row of Figure 3 (i.e., the full game setting), one immediately notices that while the XP performance all architectures except Action-In lack an interpretable pattern. Most of the architectures are at chance. However, the XP matrix for the Action-In model splits into two clusters. Within the clusters, agents show XP performance almost identical to SP, implying that they coordinate nearly perfectly with other agents trained with a different seed, whereas outside the clusters they achieve a return close to 0. As we will show in the next section, the upper cluster that has higher average XP score corresponds to the strategy where agents maximize the "similarity" between the target card and the hint card, as well as between the hint card and the guess card. In the lower, second cluster, agents try to do the opposite. They try to hint/guess cards that ideally share no common feature with the target/hint cards. In the rest of the paper, we will refer to the cluster where agents try to maximize the similarity between cards as Cluster 1, and the cluster where agents try to maximize the dissimilarity as Cluster 2. However, as we shall see in section 8.3, the Action-In agents do not just maximize/minimize feature similarity; they also demonstrate more sophisticated coordination patterns that exploit mutual exclusivity and implicature.

In the bottom row of Figure 3 (i.e., the same hand condition), DQN and RDQN also have better scores than they are in the full game. This is likely because in the simplified setting, an optimal solution that always guarantees reward exists, and therefore the agents do not need to take advantage of features to converge to that solution. Action-In still splits into two clusters; cluster 1 (hint "similar") has perfect scores for SP and XP since it is the optimal solution, and cluster 2 (hint "disimilar") is sub-optimal as agents may not tie-break when there are multiple cards the can hint/guess.

The top rows of Table 1 summarizes XP and SP scores across model architectures. We see that DQN and RDQN do not take explicit advantage of shared action and observation features which results in agents that perform well in SP but have XP scores close to chance. As we shall show shortly, they exploit features arbitrarily, resulting in miscoordination in XP. The Obs agent behaves similarly to the DQN/RDQN agents. The Action-Out and Action-In agents, on the other hand, have lower SP scores that are close to their XP scores. This is because they learn simpler conventions using common features in the observations and actions. In other words, they achieve a SP-XP trade-off using the inherent inductive bias in the attention architecture.

| Method | Full Game | | Same Hand Condition | |
| --- | --- | --- | --- | --- |
| | Cross-Play | Self-Play | Cross-Play | Self-Play |
| DQN | $0.27 \pm 0.13$ | $0.85 \pm 0.02$ | $0.79 \pm 0.09$ | $0.98 \pm 0.08$ |
| RDQN | $0.30 \pm 0.15$ | $0.86 \pm 0.02$ | $0.68 \pm 0.10$ | $0.98 \pm 0.03$ |
| Obs | $0.27 \pm 0.12$ | $0.87 \pm 0.03$ | $0.29 \pm 0.14$ | $0.92 \pm 0.03$ |
| Action-Out | $0.26 \pm 0.10$ | $0.76 \pm 0.05$ | $0.27 \pm 0.16$ | $0.82 \pm 0.08$ |
| Action-In | $0.37 \pm 0.35$ | $0.76 \pm 0.05$ | $0.45 \pm 0.47$ | $0.90 \pm 0.13$ |
| Action-In (Cluster 1) | $0.77 \pm 0.02$ | $0.82 \pm 0.02$ | $1.00 \pm 0.00$ | $1.00 \pm 0.00$ |
| Action-In (Cluster 2) | $0.71 \pm 0.03$ | $0.72 \pm 0.02$ | $0.73 \pm 0.03$ | $0.73 \pm 0.01$ |
| OP | $0.35 \pm 0.02$ | $0.35 \pm 0.02$ | $0.84 \pm 0.02$ | $0.95 \pm 0.01$ |
| OBL (level 1) | $0.27 \pm 0.05$ | $0.29 \pm 0.06$ | $0.30 \pm 0.05$ | $0.29 \pm 0.05$ |
| OBL (level 2) | $0.28 \pm 0.04$ | $0.28 \pm 0.05$ | $0.27 \pm 0.05$ | $0.29 \pm 0.05$ |

Table 1: Cross-play performance. We show results for both the full hint-guess game and the same hand condition. Each entry is the average performance is of 20 pairs of agents that are trained with different random seeds. Please refer to Fig. 3 for visualization of performance for each individual pair. Cross-Play score is the nondiagonal mean of each grid. Self-Play score is the diagonal mean, i.e. the score attained when agents play with the peer they are trained with.

## 8.2 COMPARISON WITH ZSC BASELINES.

In the bottom rows of Table 1 we also show the SP and XP results for two recent baseline ZSC methods Other-Play (OP) (Hu et al., 2020) and Off-Belief Learning (OBL) (Hu et al., 2021). For details and implementation of the baselines, see Appendix. A.3.

We see that the XP scores for OP agents only show marginal improvement over DQN agents. By preventing arbitrary symmetry breaking, OP improves XP performance, but only to a limited extent. In contrast, the OBL agents fail to obtain scores beyond chance. This is expected as OBL explicitly avoids cheap talk, which is the key for coordination in the hint-guess game.

## 8.3 POLICY EVALUATION.

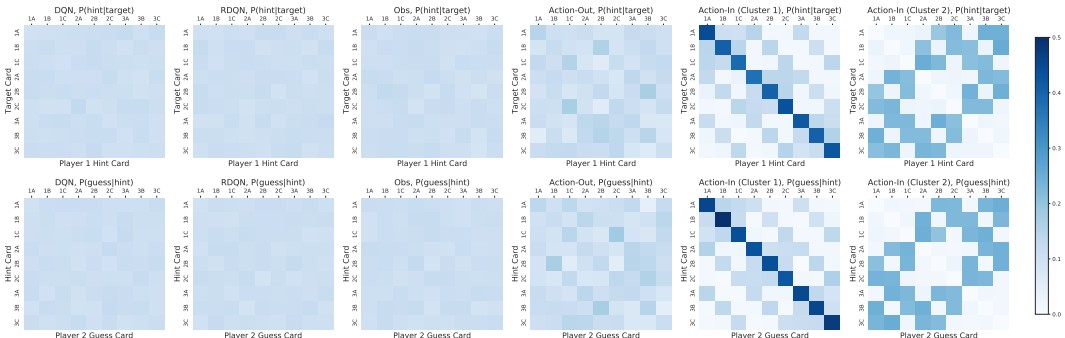

Figure 4: Policy Evaluation. The top row shows the conditional probability for the *Hinter* to hint a particular card (x-axis) when the target card is the card on the y-axis. The bottom row is the conditional probability matrix for the *Guesser* to guess a particular card (x-axis) when the hinted card is the card on the y-axis. Each subplot is the sample average of 20 agent-pairs with different seeds for 1K games within pair.

Next, we evaluate the policies of agents in more detail. In Figure 4, we provide the conditional probability for the *Hinter* to hint a card given the target card it sees (top row) and the conditional probability for the *Guesser* to guess a card given the hint card it sees (bottom row).

We analyze whether agents assign different probabilities to actions based on the features they share with the observation. We see that for DQN, RDQN and Obs, the probability matrices for both target-hint and hint-guess are nearly uniform. The probability for the *Hinter* to hint $1A$ is the same no matter the target card is exactly $1A$, or $3C$ that shares no feature in common with $1A$ across seeds. This implies that the SP policies across seeds each form their own private language for arbitrary and undecipherable coordination.

In contrast, for Action-Out, the probability matrix is visually less uniform (although still quite noisy). Indeed, in the target-hint matrix we can see that the *Hinter* will tend to hint $1A$ if it has $1A$, $2B$ if it has $1B$, $1C$ if it has $1C$, etc. In sum, Action-Out agents relatively prefer actions that are more "similar" to the observation.

For the two clusters of the Action-In model this tendency is much stronger. For cluster 1 we see that both the *Hinter* and the *Guesser* prioritize acting with the card that is exact match of the target/hint. If the exact match is not available, they turn to cards that share one feature in common. Hence, cluster 1 agents are solving $\max \sum_{i \in \{1,2\}} \mathbb{I}(f_i = \hat{f}_i)$, where $\mathbb{I}$ is the indicator function, $f_i$ / $\hat{f}_i$ are the two observation and action features. For cluster 2 we see that agents do exactly the opposite — they solve $\min \sum_{i \in \{1,2\}} \mathbb{I}(f_i = \hat{f}_i)$, and therefore put equal probability on cards that share nothing in common with the target/hint cards. In summary, the two clusters represent two equilibria Action-In agents converge to in this game — maximizing or minimizing the feature-similarity between the observation and the action.

## 8.4 SCENARIO ANALYSIS

We move on to examine in a fine-grained way whether the agents in the two clusters of the Action-In models exhibit more sophisticated behavior beyond simply minimizing/maximizing the distance between features. Here we run simulations on the four scenarios shown in Figure 2 and described in Section 5. We find that the agents in cluster 1 of Action-In demonstrate coordination patterns that are nearly identical to the human-interpretable response. These results are surprising given that our models have never been trained with any human data. Furthermore, mutual exclusivity was thought

| Scenario | Self-Play (cluster 1) | | Cross-Play (cluster 1) | | Self-Play (cluster 2) | | Cross-Play (cluster 2) | |
|---|---|---|---|---|---|---|---|---|
| | % Human | % Win | % Human | % Win | % Human | % Win | % Human | % Win |
| Exact match | 100.0% | 100.0% | 100.0% | 100.0% | 0.0% | 100.0% | 0.0% | 100.0% |
| Feature similarity | 100.0% | 100.0% | 100.0% | 100.0% | 0.0% | 100.0% | 0.0% | 100.0% |
| Mutual exclusivity | 100.0% | 100.0% | 100.0% | 100.0% | 9.3% | 91.2% | 9.3% | 92.2% |
| Similarity + Exclusivity | 92.0% | 91.7% | 97.9% | 99.5% | 3.2% | 98.4% | 0.0% | 99.9% |

Table 2: Behavioral analysis for the Action-In model in the Figure 2 scenarios. We randomly chose 10 agent-pairs from each cluster and simulated the same scenario 1K times. % Human denotes the fraction of games where the *Hinter* hints the card that corresponds to human interpretable choice (highlighted in yellow in Fig. 2), and % Win denotes the fraction of games where the *Guesser* correctly guesses.

to be hard for deep learning models to learn (Gandhi & Lake, 2020). In contrast, agents in cluster 2 always perform actions that are the opposite to the human-interpretable policy.

## 8.5 HUMAN-BASED EXPERIMENTS

We recruited 10 university students to play the hint-guess game. Each subject played as *Hinter* for 15 randomly generated games, totaling 150 different games. These subjects are then cross-matched to play as *Guessers* with the hints they generated. The human hints are also fed into randomly chosen *Guesser* agents from Action-In Cluster 1 and DQN models to test AI performance against human partners. Throughout the experiment, measures are taken to ensure that agents play zero-shot, and further details of the experiment design are available in Appendix A.2.

Human/human (*Hinter*/*Guesser*) obtained an average score of 0.75 (s.e. 0.04), and human/Action-In obtained an average score of 0.77 (s.e. 0.05). In contrast, the DQN agents obtained a score of 0.30 (s.e. 0.03). In 121 (80.7%) of the 150 games, Action-In *Guessers* chose the same guesses as human guessers, whereas in only 40.7% of the time the DQN *Guessers* agreed with humans. In summary, the Action-In *Guessers* perform comparable to human *Guessers* when given the same human-generated hints, and in most cases, both types agree on the same guesses.

## 9 CONCLUSION AND FURTHER RESEARCH

We investigated the effect of network architectures on the ability of learning algorithms to make use of the semantic relationships between shared features across actions and observations in the context of zero-shot coordination. We compared the behavior of agents with feedforward, recurrent, and attention-based architectures and find that attention-based architectures that jointly process a featurized representation of the observation and the action have a better inductive bias for exploiting this relationship. This results in high cross-play performance as well as human-compatible policies in our novel hint-guess game, which we introduced as a new test bed for coordination.

Our work points to a number of exciting future research directions. First, while we have focused on the hint-guess game, which is a specifically designed challenge for exploiting features for coordination, our attention-based method can also be tested with humans in more complex settings, e.g. in the benchmark games Hanabi or Overcooked (Carroll et al., 2019). Second, we currently assume that action features are provided as part of the problem setting. However, we believe that it would be interesting for future work to learn to discover these features from high dimensional inputs, such as images and texts. Third, humans are able to use similarity across different modalities (vision and phonetic sound in the "Bouba" and "Kiki" example) using abstract representations of learned features, and how AI can do this represents another direction for future work. Finally, we observe that, as an emergent behavior, the Action-In agents form two clusters and it would be interesting for future work to investigate the mechanisms behind cluster formation and whether clusters with different behavioral patterns would emerge in games with different dynamics.

To summarize, we believe our work shows early, intriguing results and also opens a number of exciting directions for future research to be explored by the community.

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

# A    APPENDIX

## A.1    TRAINING AND MODEL ARCHITECTURE DETAILS

**Training Setup.** We use standard experience-replay with a replay memory of size 300K. For optimization, we use the mean squared error loss, stochastic gradient descent with learning rate set to $10^{-4}$ and minibatches of size 500 each. To allow more data to be collected between training steps, we update the network only after we receive 500 new observations rather than after every observation. We use the standard exponential decay scheme with exploration rate $\epsilon = \epsilon_m + (\epsilon_0 - \epsilon_m)\exp(-n/K)$, where $n$ is the number of episodes, $\epsilon_m = 0.01$, $\epsilon_0 = 0.95$, and $K = 50,000$. All experiments were run on two computing nodes with 256GB of memory and a 28-Core Intel 2.4GHz CPU. A single training run takes roughly 8 hours for the Action-In model and 2 hours for all other models. All *Hinters* and *Guessers* have the same model structure.

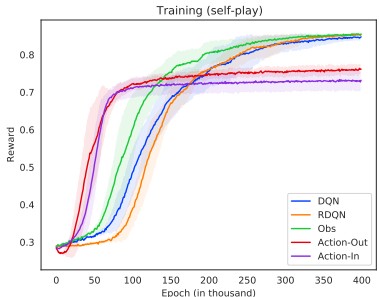

Figure 5: Training performance of different architectures for the hint-guess game. The x-axis is the number of episodes and y-axis is the mean training reward. Solid line is the mean, shading is the standard error of the mean (s.e.m) across 20 different seeds.

The model architecture details are as follows:

**Deep Q-Learning (DQN).** The DQN agent is a feedforward ReLU NN with 3 hidden layers (width 128).

**Recurrent Deep Q-Learning (RDQN).** The RDQN agent has an LSTM layer with 128 LSTM units followed by two fully-connected layers with 128 ReLU units. The input of the RDQN agent is the same as that of the DQN agent but vectors representing the cards are fed in sequentially to the LSTM before we concatenate all hidden states from each step and use that as input to a feedforward NN.

**Attention Models.** All three attention-based models share the same single-head attention layer with size determined by the shapes of the input and output. We do not add position encoding. In Obs, we add a feedforward ReLU NN with 3 hidden layers (width 128) after the attention layer. In Action-Out and Action-In, we do not add extra ReLU layers.

## A.2 DESIGN OF HUMAN-BASED EXPERIMENTS

In this section we provide details of the design of human-based experiments.

We recruited 10 individuals who are undergraduate and master students at a university. The instructions they received include: (i) rules of the hint-guess game; (ii) that they need to assume they are playing against a *Guesser* who is an ordinary human (they are not told that they would play against AI bots); (iii) that the position of the cards are permuted so they cannot provide/interpret hints based on card position, and (iv) if two cards have the same features they are effectively the same.

After they showed understanding of the instructions, the subjects were asked to act as *Hinters*. Each subject was provided with 15 randomly generated games with $F_1 = \{1, 2, 3\}$, $F_2 = \{A, B, C\}$ and $N = 5$. They were presented with both hands and the target card and were asked to write down their hints. A sample question they received was:

```
Playable card is:  2B
Your (hinter) hand is:  1C, 1B, 3B, 2C, 1A
Guesser hand is:  1B, 2C, 2B, 2C, 2B
Please choose a hint card from YOUR OWN hand!
```

After the subjects provided hints, they were given no feedback to ensure that they could not learn about the agent they are playing with; this would ensure that their actions are zero-shot.

Then we used the hints provided by the subjects to obtain human/human (*Hinter/Guesser*) and human/AI scores. To obtain human/human scores, we randomly mix-matched the subjects so that each subject would now act as the *Guesser* for 15 games generated by another human subject. This time, they were provided with both hands and the human hint, and were ask to choose one card to play. Same as before, they did not receive any feedback about their guesses to ensure zero-shot.

To obtain human/Action-In and human/DQN scores, we reproduced the 150 games and fed human hints to randomly sampled Action-In/DQN *Guesser* agents.

## A.3 DETAILS OF ZSC BASELINES

In this section we provide implementation details of Other-Play (Hu et al., 2020) and Off-Belief Learning (Hu et al., 2021), two recent zero-shot coordination (ZSC) methods that we use as baselines.

**Other-Play (OP).** The goal of OP is to find a strategy that is robust to partners breaking symmetries in different ways. To achieve this, it uses reinforcement learning to maximize returns when each agent is matched with agents playing the same policy, but under a random relabeling of states and actions under known symmetries of the Dec-POMDP (Hu et al., 2020). To apply OP, we change the objective function of the hinter from the standard self-play (SP) learning rule objective

$$\pi^* = \arg\max_{\pi} J\left(\pi^1, \pi^2\right) \tag{1}$$

To the OP objective

$$\pi^* = \arg\max_{\pi} \mathbb{E}_{\phi \sim \Phi} J\left(\pi^1, \phi\left(\pi^2\right)\right) \tag{2}$$

where the expectation is taken with respect to a uniform distribution on $\Phi$, where $\Phi$ describes the symmetries in the underlying Dec-POMDP (in the hint-and-play games, the symmetries are the two features).

As OP only changes the objective function, it can be applied on top of any SP algorithm. We choose to apply OP on top of the DQN architecture, using the same training method as detailed in Appendix. A.1, and change the objective to the OP objective. In implementation, this means that the guesser will receive a feature-permuted version of the game, i.e. feature 1 and feature 2 (the letters and the numbers) of the hands and the hint that the guesser receives will be a permuted version of what the hinter originally receives.

**Off-Belief Learning (OBL).** OBL (Hu et al., 2021) regularizes agents' ability to make inferences based on the behavior of others by forcing the agents to optimize their policy $\pi_1$ assuming past actions were taken by a given fixed policy $\pi_0$, while in the same time assuming that future actions will be taken by $\pi_1$. In practice, OBL can be iterated in a hierarchical order, where the optimal policy from the lower level becomes the input to the next higher level.

We apply OBL on DQN agents and keep the training setup the same as in Appendix A.1. At the lowest level (level 1), OBL agents assume $\pi_0$ is the policies where actions are chosen uniformly at random. And OBL level 2 assumes the policy from OBL level 1 is the new $\pi_0$, and so forth.

## A.4 EFFECT OF HAND SIZE ON PERFORMANCE

In this section we analyze the effect of hand size on agents' self-play and cross-play performance. We fix the features to be $F_1 = \{1, 2, 3\}$ and $F_2 = \{A, B, C\}$ and run experiments with hand size $N = \{3, 7\}$, as opposed to $N = 5$ in the main text. As shown in the following Table 3, SP and XP results largely confirm our findings for $N = 5$.

| Method | Full Game ($N = 3$) | | Same Hand Condition | |
| --- | --- | --- | --- | --- |
| | Cross-Play | Self-Play | Cross-Play | Self-Play |
| DQN | $0.47 \pm 0.19$ | $0.92 \pm 0.01$ | $0.87 \pm 0.11$ | $0.98 \pm 0.04$ |
| RDQN | $0.47 \pm 0.20$ | $0.92 \pm 0.01$ | $0.70 \pm 0.12$ | $0.99 \pm 0.01$ |
| Obs | $0.47 \pm 0.17$ | $0.92 \pm 0.01$ | $0.47 \pm 0.16$ | $0.95 \pm 0.03$ |
| Action-Out | $0.38 \pm 0.14$ | $0.81 \pm 0.01$ | $0.46 \pm 0.17$ | $0.88 \pm 0.12$ |
| Action-In | $0.43 \pm 0.36$ | $0.80 \pm 0.01$ | $0.45 \pm 0.47$ | $0.95 \pm 0.07$ |
| Action-In (Cluster 1) | $0.81 \pm 0.02$ | $0.81 \pm 0.02$ | $1.00 \pm 0.00$ | $1.00 \pm 0.00$ |
| Action-In (Cluster 2) | $0.80 \pm 0.01$ | $0.80 \pm 0.01$ | $0.89 \pm 0.06$ | $0.90 \pm 0.02$ |
| OP | $0.55 \pm 0.03$ | $0.54 \pm 0.04$ | $0.90 \pm 0.01$ | $0.98 \pm 0.01$ |
| OBL (level 1) | $0.47 \pm 0.07$ | $0.44 \pm 0.06$ | $0.47 \pm 0.03$ | $0.48 \pm 0.03$ |
| OBL (level 2) | $0.47 \pm 0.04$ | $0.46 \pm 0.06$ | $0.47 \pm 0.05$ | $0.47 \pm 0.05$ |

| Method | Full Game ($N = 7$) | | Same Hand Condition | |
| --- | --- | --- | --- | --- |
| | Cross-Play | Self-Play | Cross-Play | Self-Play |
| DQN | $0.22 \pm 0.08$ | $0.72 \pm 0.03$ | $0.77 \pm 0.10$ | $0.96 \pm 0.08$ |
| RDQN | $0.25 \pm 0.09$ | $0.76 \pm 0.01$ | $0.62 \pm 0.14$ | $0.95 \pm 0.05$ |
| Obs | $0.24 \pm 0.09$ | $0.77 \pm 0.03$ | $0.24 \pm 0.08$ | $0.93 \pm 0.01$ |
| Action-Out | $0.22 \pm 0.11$ | $0.77 \pm 0.02$ | $0.22 \pm 0.13$ | $0.80 \pm 0.09$ |
| Action-In | $0.37 \pm 0.33$ | $0.70 \pm 0.06$ | $0.40 \pm 0.38$ | $0.89 \pm 0.11$ |
| Action-In (Cluster 1) | $0.78 \pm 0.01$ | $0.78 \pm 0.03$ | $1.00 \pm 0.00$ | $1.00 \pm 0.00$ |
| Action-In (Cluster 2) | $0.67 \pm 0.02$ | $0.67 \pm 0.02$ | $0.67 \pm 0.03$ | $0.67 \pm 0.02$ |
| OP | $0.32 \pm 0.02$ | $0.31 \pm 0.02$ | $0.79 \pm 0.02$ | $0.95 \pm 0.02$ |
| OBL (level 1) | $0.21 \pm 0.04$ | $0.22 \pm 0.03$ | $0.22 \pm 0.03$ | $0.22 \pm 0.04$ |
| OBL (level 2) | $0.24 \pm 0.03$ | $0.22 \pm 0.03$ | $0.22 \pm 0.03$ | $0.25 \pm 0.02$ |

Table 3: Cross-play performance for card number $N = 3$ and $N = 7$. We show results for both the full hint-guess game and the simplified version where both agents have the same hand. Each entry is the average performance is of 20 pairs of agents that are trained with different random seeds. Cross-Play score is the nondiagonal mean of each grid. Self-Play score is the diagonal mean, i.e. the score attained when agents play with the peer they are trained with.

