# OpenReview forum: "Zero-Shot Coordination via Semantic Relationships Between Actions and Observations"
_ICLR.cc/2022/Conference — ICLR 2022 Submitted_

### Official Review · Reviewer_JS7v · 2021-10-31

**Correctness:** 3
**Technical Novelty And Significance:** 3
**Empirical Novelty And Significance:** 2
**Recommendation:** 6
**Confidence:** 2

**Main Review:**


The paper shows that in the case of the proposed new game, that
"attention-based architectures that jointly process a
featurized representation of the observation and the action" performed
better. The evidence shown supports this claim.

This paper is quite far from my expertise, and there are a number of
confusions I had about the paper:
- What is special about the new hint-guess game? Does it feature some aspect that
  the games used in current experiments ignore?
- Why is it called zero-shot coordination, when there is considerable training
  using training data that seems virtually identical to the proposed task?
- What can we generalize from the game/experiments that is valuable for other
  tasks? In particular, given the goal of having 'human-compatible' learning,
  how is this addressed?
- Figure 5 shows the  probability of proposed 'hints' given
all possible target 'answers'. Is is surprising that they are largely uniform, given that any
possible card could be the appropriate answer depending on the configuration of the other
cards?

- In Fig 5, I don't see the connection between the card-selection probabilities
and the conclusion of 'private languages'. They seem to be 2 different questions and
I don't see how the results in Fig 5 connects them.
- Section 8.3 says: "We find that the agents in cluster 1 of Action-In demonstrate
coordination patterns that are nearly identical to the human-interpretable response."
There are no human-based experiments, and I'm unclear what the evidence is that supports
this claim even if there was some human data.
- The final discussion says "relies on a majority vote". I did not see any mention of
a voting mechanism previously.
- Should the first formula on page 6 be S_k = instead S_i =
- Do any of the observations from the analysis of the Hint-Guess game apply to other
collaboration games?
- Why were 5 cards chosen as the configuration? Do results vary with other numbers?
- What is the intuition behind each proposed attention mechanism?


**Summary Of The Paper:**

This paper explores several architectures for reinforcement-learning in the
context of a newly proposed card game (Hinter-Guesser).
A number of the architectures seem to have poor performance on the game.
Moreover, whatever they learn is not compatible with what other learners learn,
so that they cannot mutually understand Hints & Guesses. The stated aim is:
"to build an agent with the capacity to develop these kinds of abstract correspondences during
self-play such that they can robustly succeed during cross-play or during play with people."
The architectures generally succeed (70-90%) at playing against themselves (with the shared learned
policy) but perform badly when playing with independently trained agents,
except when trained with the desired action as part of the input (unclear how
this can be tested on unknown cases when 'answer' is given as part of the
inputs).



**Summary Of The Review:**

The proposed approach works for this bespoke game. It would be good to have more evidence that the approach is generally useful.

---

> ### Author Response · Authors · 2021-11-15
> **Response to Reviewer JS7v (part 1)**
>
> Many thanks for your insightful review and questions!
>
> Frist, in response to comments and suggestions from reviewers, **we have made significant additions and changes to the paper**, including human experiments and comparison with ZSC baselines, and details can be found at the Overall Response above (https://openreview.net/forum?id=j97zf-nLhC&noteId=mi1ro3GHjv2).
>
> Also, we try to address your questions and concerns below:
>
> ***
>
> > **Q1** “What is special about the new hint-guess game? Does it feature some aspect that the games used in current experiments ignore?”
>
> Reply: The hint-and-guess includes features that can be used for coordination. Features are also present in other games that are challenging from a coordination point of view, e.g. Hanabi, but those games contain a large number of other challenges. In contrast, our hint-guess game is entirely focused on this aspect of coordination, which has thus far been largely neglected by the community.
>
> > **Q2** “Why is it called zero-shot coordination, when there is considerable training using training data that seems virtually identical to the proposed task?”
>
> Reply: As described in Section 2 & 3, we follow the terminology from prior work which formally introduced the Zero-Shot *Coordination* setting. This is not about the number of samples used in the environment, but about the fact that coordination success is measured without any shared experience (“zero-shot”). The goal of ZSC is to develop algorithms that allow independently trained agents to coordinate at test time.
>
> > **Q3** “What can we generalize from the game/experiments that is valuable for other tasks? In particular, given the goal of having 'human-compatible' learning, how is this addressed?”
>
> Reply: A potential next-step is to test our model architecture in more complicated games, such as Hanabi. We also expect these techniques to be valuable in human-robot interactive settings where agents might point in a spatially grounded way or take other actions that have features associated with them.
>
> > **Q4** “Figure 5 shows the probability of proposed 'hints' given all possible target 'answers'. Is it surprising that they are largely uniform, given that any possible card could be the appropriate answer depending on the configuration of the other cards?”
>
> Reply: It depends what is meant by the word “appropriate”. If the reviewer means appropriate in the sense of maximizing the expected return, we agree---it is not surprising that there may be complex relationships between the configuration of the other cards and the appropriate hint. However, our interest here is not only in maximizing expected return, but in regularizing learning toward “good” but “human interpretable” policies. In this latter sense, the “appropriate” may be the card that is most semantically similar to the target card, as demonstrated by cluster 1.
>
> **P.S. In the updated version of the paper, what used to be Figure 5 is now Figure 4.**
>
> > **Q5** “In Fig 5, I don't see the connection between the card-selection probabilities and the conclusion of 'private languages'. They seem to be 2 different questions and I don't see how the results in Fig 5 connect them.”
>
> Reply: The fact that there is no semantic relationship between the ‘hints’ and ‘targets’ implies that the agents use highly *arbitrary* mappings to communicate information, which are inconsistent across different pairs. This is further supported by the low cross-play scores.
>
> **Response continues to the next post...**

---

> > ### Author Response · Authors · 2021-11-15
> > **Response to Reviewer JS7v (part 2)**
> >
> > > **Q6** “There are no human-based experiments, and I'm unclear what the evidence is that supports this claim even if there was some human data.”
> >
> > Reply: In the updated version, we have added a section (sec. 8.5) where we present human-based experiment results, comparing human-human and human-AI performance, across DQN and Action-In. We find that the Action-In models accomplish human-AI coordination comparable to human-human performance while the baseline DQN does not. More specifically, human/human (Hinter/Guesser) achieved an average score of 0.75 (s.e. 0.04), and human/Action-In achieved an average score of 0.77 (s.e. 0.05). In contrast, the DQN Guessers obtained a score of 0.30 (s.e. 0.03). In 121 (81%) of the 150 games, Action-In Guessers chose the same guesses as human Guessers, whereas in only 41% of the time the DQN Guessers agreed with humans.
> >
> > > **Q7** “The final discussion says "relies on a majority vote". I did not see any mention of a voting mechanism previously.”
> >
> > Reply: Sorry for unclear wording here; we have revised wording of this phrase in the updated manuscript. What we actually meant was to choose the clusters based on within-cluster scores, as different equilibria will have different expected returns. In the hint-and-play game in particular, Action-In cluster 1 (which had a policy intuitive to humans) has higher within-cluster scores than cluster 2 (which was opposite the intuitive policy), and this is apparent in the same hand condition, where cluster 1 has a perfect XP score of 1.
> >
> > > **Q8** “Do any of the observations from the analysis of the Hint-Guess game apply to other collaboration games?”
> >
> > Reply: We think so! First, human intelligence is fundamentally embodied in the real world, which contains features and agents that act on objects. As such, these results are important for coordination in the physically grounded world, beyond the scope of abstract card games. Second, although we have not investigated explicit communication systems such as natural language -- the starting place for linguistic communication is coordination and we hope our work will influence those studying natural language and NLP to consider words as having features in and of themselves (e.g., the Bouba/Kiki example in the introduction).
> >
> > > **Q9** “Why were 5 cards chosen as the configuration? Do results vary with other numbers?”
> >
> > Reply:  The results are qualitatively the same with more or less cards. We have now added evaluation of self-play and cross-play performance with N=3 and N=7 in the Appendix (A.4). Note, that the results largely confirm our findings for N=5.
> >
> > > **Q10** “What is the intuition behind each proposed attention mechanism?”
> >
> > Reply:  Each of the attention mechanisms we investigate let’s us probe one specific ablation compared to the final model. The intention is to better understand which component(s) of the final model were causal of the enhanced performance in zero-shot coordination. Our conclusion is that the attention architecture needs to take both the action and observations as input to allow the agent to fully exploit the semantic relationships for coordination.
> >
> > ***
> >
> > **In summary, we hope that with our explanations, revisions, and new results, our points have been made clearer and stronger. Thank you again for your insightful questions and comments!**

---

> > > ### Author Response · Authors · 2021-11-28
> > > **Follow up from Paper3936 Authors**
> > >
> > > Dear Reviewer,
> > >
> > > We again thank you for taking your precious time to provide an insightful review. We would like to follow up to see if the reviewer had any more questions/feedback. In particular, we would like to get feedback on:
> > >
> > > - Is the reviewer in agreement on the clarification above regarding the reviewer's questions?
> > > - Does the reviewer agree that our addition of new results (baselines, human experiments, robustness tests, etc.) and revisions have made our approach more convincing?
> > >
> > > If the answer is yes to the above questions, would the reviewer be willing to consider raising their score?
> > >
> > > If no, would the reviewer be willing to engage in further discussion?
> > >
> > > Thank you again for your thoughtful review and comments!
> > >
> > > Best, ICLR 2022 Conference Paper3936 Authors

---

> > > > ### Comment · Reviewer_JS7v · 2021-11-28
> > > > **revised score**
> > > >
> > > > Based on the authors reply, the expertise and opinions of the other reviewers, and the improvements to the paper, I'm happy to revise my rating to 'marginally above'. I don't have  enough expertise in this topic to support a higher rating. Maybe some of the other reviewers can consider this.

---

### Official Review · Reviewer_CXsY · 2021-11-01

**Correctness:** 4
**Technical Novelty And Significance:** 3
**Empirical Novelty And Significance:** 3
**Recommendation:** 6
**Confidence:** 4

**Main Review:**

Strengths:
* The introduction is nicely written and introduces ample motivation from multiple backgrounds.
* The proposed task is very clearly described, as well as scenarios of interests / challenges.
* Experiment visualization is very informative, showcasing interesting dichotomy of self-play vs cross-play performances.
* The experiments seem to have enough details and reproducible.

Weakness:
* I feel like section 3 and 4 a bit verbose. Even for me, someone not in the specific field of multi-agent RL, the text seems a bit redundant.
* Notation in section 4 is not really clear, with some notation overloading / ambiguity. For example, the number of players n is overloaded with each object's number of feature n.
* No baseline in ZSC is considered. In related work, it's mentioned that prior work needs some experimenter-coded input. But I don't see why similar baseline can't be examined here, either as a baseline or an upper-bound.
* The motivation behind having same-hand setup is unclear to me.

**Summary Of The Paper:**

The paper introduces a new Zero-shot Coordination (ZSC) task which involves a Hinter and a Guesser and a set of explicit features shared between the two. The paper evaluates several methods and conclude that an attention mechanism that put attention on both observation and action features is helpful to produce agents that are successful in cross play.

**Summary Of The Review:**

Though I'm not an expert in multi-agent RL, I think the work will be interesting to the field.
The paper offers an interesting study showing that action and observation information should both be leveraged to form interesting communication / coordination strategy.
However, the paper is limited by the features are explicitly available, making strategy generation less impressive.

---

> ### Author Response · Authors · 2021-11-15
> **Response to Reviewer CXsY**
>
> Many thanks for your insightful comments and thorough review.
>
> Frist, to address comments and suggestions from reviewers, **we have made significant additions and changes to the paper**, and details can be found at the Overall Response above (https://openreview.net/forum?id=j97zf-nLhC&noteId=mi1ro3GHjv2).
>
> Also, in what follows we will try to address your concerns about the weaknesses of our paper:
>
> ***
>
> > **Concern 1** “I feel like section 3 and 4 a bit verbose. Notation in section 4 is not really clear”
>
> Reply: Thank you for your suggestion. In response to your suggestion, we have substantially improved clarity and succinctness in our updated version for sections 3 and 4.
>
> > **Concern 2** “No baseline in ZSC is considered.”
>
> Reply: We added (sec. 8.2), a comparison of self-play (SP) and cross-play (XP) performance of our methods to two recent zero-shot coordination baselines: other-play (OP) and off-belief learning (OBL). Our results show that both OP and OBL underperform Action-In in our setting. In XP, OP obtained 0.35 (s.e. 0.02) and OBL (level 2) obtained 0.28 (s.e. 0.04). By preventing arbitrary symmetry breaking, OP improves XP performance compared with DQN (0.27±0.13), but to a limited extent. In contrast, the OBL agents fail to obtain scores beyond chance either in SP or XP. This is expected as OBL explicitly avoids cheap talk, which is crucial for coordination in the hint-guess game (as well as in many other settings).
>
> > **Concern 3** “The motivation behind having same-hand setup is unclear to me.”
>
> Reply: The same hand condition let’s us study the methods, baselines and ablations in a simplified setting where there is an *optimal policy* (for both self-play and ZSC) that is very simple (always hint and play the target card). In contrast, in the full setting there is a trade-off between coordination (XP) and self-play performance.
>
> > **Concern 4** “The paper is limited by the features are explicitly available”
>
> Reply: This is a great point. We currently assume that action features are provided as part of the problem setting. However, we believe that it would be interesting for future work to learn to discover these features from high dimensional inputs, such as images and texts. We have revised the Further Research section and add a discussion related to this point to incorporate your insight. Still, we believe our work is the first that shows early, intriguing results for using attention to exploit semantic relationships between actions and features in a Dec-POMDP setting and it opens many further research possibilities.
>
> ***
>
> **In summary, we hope that with our explanations, revisions, and new results, our points have been made clearer and stronger. Thank you again for your insightful questions and comments!**

---

> > ### Author Response · Authors · 2021-11-29
> > **Follow up from Paper3936 Authors**
> >
> > Dear Reviewer,
> >
> > We again thank you for taking your precious time to provide an insightful review. We would like to follow up to see if the reviewer had any more questions/feedback. In particular, we would like to get feedback on:
> >
> > - Is the reviewer in agreement on the clarification above regarding the reviewer's concerns?
> > - Does the reviewer agree that our addition of new results (baselines, human experiments, robustness tests, etc.) and revisions have made our approach more convincing?
> >
> > If the answer is yes to the above questions, would the reviewer be willing to consider raising their score?
> >
> > If no, would the reviewer be willing to engage in further discussion?
> >
> > Thank you again for your thoughtful review and comments!
> >
> > Best, ICLR 2022 Conference Paper3936 Authors

---

### Official Review · Reviewer_Ba9g · 2021-11-02

**Correctness:** 3
**Technical Novelty And Significance:** 3
**Empirical Novelty And Significance:** 4
**Recommendation:** 6
**Confidence:** 3

**Main Review:**

Strengths:
1) The paper is well organized. The idea and method are clearly presented. The experiments are conducted in a novel environment (proposed Hint-and-Guess game).
2) The paper proposes a novel method to extend Dec-POMDP with shared feature representations for both observations and actions.
3) The paper develops a novel Hint-and-Guess game for evaluating effects of different deep learning architectures.

Weaknesses:
1) Some details of the experiment setting may need further explanation. It is not clear why the paper fixes hand size N=5 (and feature size) for the game, and how different hand sizes may affect the performance.
2) More analysis of the results is needed. Why the 13 agents tend to make two clusters with two groups of coordination types? Why these agents form different clusters for the two versions of full game and same hand condition?
3) What would your method work in other zero-shot coordination games other than your proposed game?

**Summary Of The Paper:**

The paper exploits semantic relationships between features of observations and an action for zero-shot coordination in multi-agent reinforcement learning. It extends decentralized partially observable Markov decision process (Dec-POMDP) by representing observations and actions with shared features. Technically it uses attention mechanism of deep learning to exploit the semantic relationships between observations and actions. Besides, the paper develops a novel human interpretable environment for analysis.

**Summary Of The Review:**

In summary, the problem tackled in the paper sounds interesting and the idea and method are reasonable and easy to follow. While more explanations and analysis are preferred for the experiments, my current rating leans to acceptance.

---

> ### Author Response · Authors · 2021-11-15
> **Response to Reviewer Ba9g**
>
> Many thanks for your insightful review and encouraging comments!
>
> Frist, to address comments and suggestions from reviewers, **we have made significant additions and changes to the paper**, including human experiments and ZSC baselines, and details can be found at the **Overall Response** above (https://openreview.net/forum?id=j97zf-nLhC&noteId=mi1ro3GHjv2).
>
> Also, in what follows we try to address your concerns about the weaknesses of our paper.
>
> ***
>
> > **Q1** “It is not clear why the paper fixes hand size N=5 (and feature size) for the game, and how different hand sizes may affect the performance.”
>
> Reply: There is nothing special about the N=5 for the hand size. To demonstrate that our results are robust for different hand sizes, we added a section in the Appendix (A.4) to demonstrate the effect of hand size on self-play and cross-play performance. The results largely confirm our findings for N=5. In terms of feature size, it is relative to hand size, in the sense that when the hand size is fixed, the more features, the lower probability that a particular card will be in the Hinter’s hand. Thus, increasing feature size is effectively decreasing the hand size, making the game harder for the Hinter.
>
> > **Q2** “Why do the 13 agents tend to make two clusters with two groups of coordination types?”
>
> Reply: There is nothing special about the number 13 in our experiments. The same would hold true for a different number of independently trained seeds. The two clusters are an emergent property of the problem setting and the semantic features. In particular, they correspond to hinting for the “nearest”  match (cluster I), where the card that is the closest in properties to the target is chosen, and the “distant match”, which is the opposite.
>
> > **Q3** “Why do these agents form different clusters for the two versions of full game and same hand condition?”
>
> Reply: The clusters are an emergent property of the optimization process. Naturally, different problem settings will give rise to different strategies. In the hint-and-guess game in particular, it is possible that two clusters emerged as they correspond to the two most intuitive strategies - hint most similar and most dissimilar. Indeed, we have included your insight in the Further Research section and suggested that it would be interesting to investigate the mechanisms behind cluster formation (e.g. relative size of clusters) and whether clusters with different behavioral patterns would emerge in games with different dynamics.
>
> > **Q4** “What would your method work in other zero-shot coordination games other than your proposed game?”
>
> Reply: Yes, our method is in principle applicable whenever action features are present and can be used for coordination purposes.
>
> ***
>
> **In summary, we hope that with our explanations, revisions, and new results, our points have been made clearer and stronger. Thank you again for your insightful questions and comments!**

---

> > ### Author Response · Authors · 2021-11-28
> > **Follow up from Paper3936 Authors**
> >
> > Dear Reviewer,
> >
> > We again thank you for taking your precious time to provide an insightful review. We would like to follow up to see if the reviewer had any more questions/feedback. In particular, we would like to get feedback on:
> >
> > - Is the reviewer in agreement on the clarification above regarding the reviewer's questions?
> > - Does the reviewer agree that our addition of new results (baselines, human experiments, robustness tests, etc.) and revisions have made our approach more convincing?
> >
> > If the answer is yes to the above questions, would the reviewer be willing to consider raising their score?
> >
> > If no, would the reviewer be willing to engage in further discussion?
> >
> > Thank you again for your thoughtful review and comments!
> >
> > Best, ICLR 2022 Conference Paper3936 Authors

---

> > > ### Comment · Reviewer_Ba9g · 2021-11-29
> > > **Keep the rating**
> > >
> > > I am almost satisfactory with the authors' reply and revision. However, I don't have enough expertise in this topic, and after reading detailed comments and raised concerns from other expertised reviewers (e.g., Reviewer rNES), I would like to keep my rating of 'marginally above'.

---

### Official Review · Reviewer_rNES · 2021-11-02

**Correctness:** 3
**Technical Novelty And Significance:** 2
**Empirical Novelty And Significance:** 3
**Recommendation:** 6
**Confidence:** 3

**Main Review:**

The paper presents a nice conceptual overview about the need for human compatibility of actions associated with observations in zero shot coordination scenarios. They list the problems that arise when statistically independent actions are assigned to observations. The resulting disconnect not only affects human interpretability, but also lacks consistency during cross-play in cooperative games such Hanabi.

The proposed approach for zero-shot coordination learns to assign actions to observations in a semantically meaningful way. To produce semantic actions, the action feature is embedded along with the observation to predict contextualized action. The method itself is not new, and is a standard technique employed in many language and vision tasks such as matching models, where the query is concatenated with the featurized responses before passing them through a classification network that predicts the responses. It is surprising that the idea has not been explored in POMDP networks before, and a more comprehensive related work section could help in understanding why this is the case.

The paper points to several logical primitives used in the game play by humans, such as exact match, similarity, implicatures such as mutual exclusivity and similarity + exclusivity. There could be several other strategies used during game-play. How many of these strategies are actually learnt by the algorithm? How did the authors find one-to-one correspondence between the human designed strategies, and the statistically learnt strategies?
A related question is how does the model choose diverse strategies at different times during the game-play. There are no regularizations on policy selection, so one can assume that the model might collapse to learn a single simple strategy e.g., exact match instead of the diverse logic plays.

Given the the action featurization is weak (just a one-hot vector), how does the method ascertain that the action features are not leaking into the contextualized policy prediction? This looks likely when we look at the clustered results from Action-In experiments, where the model shows two modes - exact similarity and exact dissimilarity. Would the model still work (or probably work better) if higher order embeddings are added to reduce the actual feature distance between input action and output?

The action-observation relationship is achieved by concatenating the features before passing them through an attention model. The paper does not explain why concatenation + attention module is an ideal choice for modeling the logic, or if any competing designs are worth exploring. Could the features be modeled differently such as within a Bayesian framework, or through deep generative models?  A few methods are mentioned in the related work, but none of them are used as a baseline technique.




**Summary Of The Paper:**

The paper proposes an approach for zero-shot coordination that learns to assign actions to observations in a semantically meaningful way. The authors show that semantic actions have a better inductive bias leading to increased consistency in cross-play. Their proposed method is a learning based approach where an attention model jointly processes featurized representation of the observation and the action. The evaluations show that the agents produce human-compatible policies on a simplified Hanabi card game play.


**Summary Of The Review:**

Strengths:
1. Topical: Looks into a relatively new problem of generalization during cross play after training using self-play.
2. Strong conceptual overview and motivation with examples.
3. Use of attention based embedding for contextualized policy prediction seems to be new in POMDPs, although it has been used in vision and NLP applications before.

Weaknesses:
1. Weak literature review with very general coverage of the problem space.
2. Lack of strong baselines - The related work mentions Bayesian/causal reasoning based approaches that are considered too expensive. It might help to show the performance-accuracy tradeoff of those methods vis-a-vis currently proposed approach.
3. The quantitative evaluations are not well explained / justified. E.g.,
  (a) why does the baseline DQN (simplest model) perform remarkably well for the same hand condition, while the proposed approach has significantly worse accuracy?
  (b) Why does the proposed approach always converge into two clusters of related cross-plays?
  (c) How is the diversity of the learnt policies measured?

Overall, although the problem itself is novel, the treatment is not convincing enough and does not meet the bar for an ICLR acceptance yet.

---

> ### Author Response · Authors · 2021-11-15
> **Response to Reviewer rNES**
>
> We sincerely thank the reviewer for an insightful review.
>
> Frist, thanks to comments and suggestions from reviewers, **we have made significant additions and changes to the paper**, including human experiments and ZSC baselines, and details can be found at the Overall Response above (https://openreview.net/forum?id=j97zf-nLhC&noteId=mi1ro3GHjv2).
>
> Next, in what follows we try to address the questions and concerns raised by the reviewer.
>
> ***
>
> ## Questions
>
>
> > **Q1** “How many of these strategies are actually learnt by the algorithm?”
>
> Reply: The model does not learn these strategies explicitly -- they simply emerge through training. The model maps the given observation and each of the actions to the expected return for said action. In order to characterize the behavior of these trained models we use specific scenarios (sections 8.4) that are characteristic of a specific class of higher order reasoning.
>
>
> > **Q2** “How did the authors find one-to-one correspondence between the human designed strategies, and the statistically learnt strategies?”
>
> Reply: We do not find a one-to-one correspondence, instead we simply report what the model does in the specific situations (sec. 8.4). That is, we initialized games in such configurations, and recorded how the models hinted/guessed. The manuscript has been updated to make this point clearer.
> We have also included human experiments to illustrate our point, which is summarized in the Global Response above.
>
> > **Q3** “How does the model choose diverse strategies at different times during the game-play. There are no regularizations on policy selection, so one can assume that the model might collapse to learn a single simple strategy e.g., exact match instead of the diverse logic plays.”
>
> Reply: The reason that the model does not collapse to exact match (for example) is that exact match cannot be applied in most scenarios. The cards that are dealt to the Hinter and Guesser are random---much of the time, the target card is not present in the Hinter’s hand, so the Hinter cannot execute an exact match strategy. As a consequence, the Action-In Cluster 1 models learn the “hint nearest” strategy. However, this strategy will also fail in some settings which, in turn, leads to the other, richer behaviours (“implicatures”) that we observe.
>
> > **Q4** “How does the method ascertain that the action features are not leaking into the contextualized policy prediction? … Would the model still work (or probably work better) if higher order embeddings are added to reduce the actual feature distance between input action and output?”
>
> Reply: We humbly request a bit more clarification here. Do you mean that we could add more layers to the attention mechanism? If so, this is an experiment we are happy to run. Otherwise, we would be appreciative if you could clarify your comment.
>
> > **Q5** “The paper does not explain why concatenation + attention module is an ideal choice for modeling the logic, or if any competing designs are worth exploring. Could the features be modeled differently such as within a Bayesian framework, or through deep generative models.”
>
> Reply: We never claim that our architecture is optimal or “ideal” for the given task. It is a well motivated design choice though, since we want to generate pairwise relational structures between observations and actions. Also, it is standard to pass concatenated features through attention based neural networks.
> We also explore a number of different options and ablations for using attention in this setting.
> Lastly, it is unclear to us how deep generative models relate to our work, it would be great if you could clarify, and we are happy to include more discussion about this.
>
> ***
>
> **Response continues to the next post...**

---

> > ### Author Response · Authors · 2021-11-15
> > **Response to Reviewer rNES (cont.)**
> >
> > **continuing from the previous post...**
> >
> > ***
> >
> > ## Concerns
> >
> > > **Concern 1** “Weak literature review with very general coverage of the problem space.”
> >
> > Reply: We have substantially revised the Related Work section to provide more specific discussions on how other work related to our paper. Also, many thanks for pointing out that we should discuss related work in other areas of deep learning that use attention models where action features are embedded in observations. In fact, we are more than happy to include a discussion about this! In the updated manuscript, we have included a discussion on question answering models in NLP and attentive neural processes for regression tasks.
> > We would also greatly appreciate it if you could point to more papers we can include for this direction.
> >
> > >  **Concern 2** “Lack of strong baselines.”
> >
> > Reply: We added (sec. 8.2), a comparison of self-play (SP) and cross-play (XP) performance of our methods to two recent zero-shot coordination baselines: other-play (OP) and off-belief learning (OBL). Our results show that both OP and OBL underperform Action-In in our setting. In XP, OP obtained 0.35 (s.e. 0.02) and OBL (level 2) obtained 0.28 (s.e. 0.04). By preventing arbitrary symmetry breaking, OP improves XP performance compared with DQN (0.27±0.13), but to a limited extent. In contrast, the OBL agents fail to obtain scores beyond chance either in SP or XP. This is expected as OBL explicitly avoids cheap talk, which is crucial for coordination in the hint-guess game (as well as in many other settings).
> >
> > > **Concern 3** “The quantitative evaluations are not well explained / justified. E.g., (a) (b) and (c)”
> >
> > **We try to address the concerns (a)-(c) one by one below:**
> >
> > > **a.** “Why does the baseline DQN (simplest model) perform remarkably well for the same hand condition, while the proposed approach has significantly worse accuracy?”
> >
> > Reply: The proposed approach (Action-In) produces two different clusters in the same hand condition. Cluster I uses a “hint the target card” strategy, whereas cluster II uses a “hint the most unlike the target card” strategy. These strategies are simple, reproducible (there is no gap between SP and XP), and produce reasonable performance (cluster I performs perfectly in XP while cluster II performs comparably with DQN XP), but they are clearly mutually incompatible. It is because of this incompatibility that the XP across all runs is much lower than the XP among each cluster. For DQN, we think it performs well because in the simplified setting, an optimal solution that always guarantees reward exists, and therefore the agents do not need to take advantage of features to converge to that solution.
> >
> > > **b.** "Why does the proposed approach always converge into two clusters of related cross-plays?”
> >
> > Reply: The formation of clusters is an emergent behavior and this is expected to be different in different settings. In the hint-and-guess game, it is possible that two clusters emerged as they correspond to the two most intuitive strategies - hint most similar and most dissimilar. We have included your insight in the Further Research section and mentioned it would be interesting to investigate the mechanisms behind cluster formation and whether clusters with different behavioral patterns would emerge in games with different dynamics.
> >
> > > **c.** “How is the diversity of the learnt policies measured?”
> >
> > Reply: We analyse the cross-play performance of the resulting policies and also inspect individual scenarios (Exact Match, Feature Similarity, Implicature) to gain further insights.
> >
> > ***
> >
> > **In summary, we hope that with our explanations, revisions, and new results, our points have been made clearer and stronger. Thank you!**

---

> > > ### Author Response · Authors · 2021-11-29
> > > **Thank you for raising the score!**
> > >
> > > Dear Reviewer,
> > >
> > > We are grateful that the reviewer decided to raise their score, and we would like to follow up to see if the reviewer had any more questions/feedback. In particular, we would like to get feedback on:
> > >
> > > - Does the reviewer feel that their concerns have been addressed properly in the current version of the manuscript?
> > > - Does the reviewer think the addition of new results (baselines, human experiments, robustness tests, etc.) made the proposed approch more convincing?
> > >
> > > Thank you again for your thoughtful review and comments!
> > >
> > > Best, ICLR 2022 Conference Paper3936 Authors

---

### Author Response · Authors · 2021-11-15
**Overall Response: Significant Additions and Revisions**

We would like to thank all reviewers for the insightful reviews and comments. In response to suggestions from the reviewers, we made a few **significant additions and changes to the paper** which we highlight here.

***

## Additions:


- **Human-based experiments**: We added a section (sec. 8.5) where we present human-based experiment results, comparing human-human and human-AI performance, across DQN and Action-In. We find that the **Action-In (cluster I) models obtain high human-AI coordination scores** comparable to human-human performance while the baseline DQN does not (see table below). Moreover, in 121 (81%) of the 150 games, Action-In Guessers chose the same guesses as human Guessers, whereas in only 41% of the time the DQN Guessers agreed with humans.
| Human-Human | Human-Action-In | Human-DQN |
|-------------|-----------------|-----------|
| 0.75±0.04   | 0.77±0.05       | 0.30±0.03 |

&emsp;&emsp;*Table: XP scores between human Hinters and given Guessers, for details see sec. 8.5 of paper*

- **ZSC-baselines**: We added (sec. 8.2), a comparison of self-play (SP) and cross-play (XP) performance of our methods to two recent zero-shot coordination baselines: other-play (OP) and off-belief learning (OBL).
Our results show that **both OP and OBL underperform Action-In** in our setting (summary table below). By preventing arbitrary symmetry breaking, OP improves XP performance compared with DQN, but to a limited extent. In contrast, the OBL agents fail to obtain scores beyond chance. This is expected as OBL explicitly avoids cheap talk, which is crucial for coordination in the hint-guess game (as well as in many other settings).
| DQN       | Action-In (Cls. I) | OBL (level 1) | OBL (level 2) | OP        |
|-----------|-----------------------|---------------|---------------|-----------|
| 0.27±0.13 | 0.77±0.02             | 0.27±0.05     | 0.28±0.04     | 0.35±0.02 |

&emsp;&emsp;*Table: XP scores for Action-In and baselines, for more details see sec. 8.2 of paper*

- **Effect of hand size**: We added a section in the Appendix (A.4) to demonstrate the effect of hand size on SP and XP performance of different model architectures and baseline methods (OP/OBL). We find that our main results hold up for different hand sizes (N=3 or N=7).

***

## Revisions:

- **Related work (sec. 2)**:  We have added a discussion about attention models that exploit semantic relationship between inputs/outputs in other areas of deep learning. We have also provided more specific discussions on how other work relate to our idea (R1).


- **Dec-POMDP with features (sec. 4)**: We have significantly improved the clarity of the notation. (R3, R4).


- **Further research (sec. 9)**: We have incorporated reviewer insights and suggestions into the further research section.


- **Other sections**: We have also revised other sections to address various suggestions from the reviewers.

Please see individual replies for detailed comments.

---

**=============================== Summary Updates 2021/11/29 ===============================**

By the end of the discussion period, we would like to thank all reviewers again for their insightful comments.

In particular, we are encouraged that the reviewers think our paper:

- studies a novel problem (R1, R3)
- proposes a novel method (R1, R2)
- is supported by experiment results (R3, R4)
- is conceptually well-organized (R1, R2, R3)

We also notice that before the discussion period, the main concerns of the reviewers were:

- lack of ZSC baselines (R1, R3)
- some unclear notation for the Dec-POMDP feature section (R3, R4)
- some empirical results needed more clarification (R1, R2, R4)
- the Action-In method has some limitations, e.g. the features need to be made explicitly available (R3, R4)

In the revised manuscript, we tried to fix these issues (see above for Additions and Revisions). For the limitations pointed out by reviewers, we have made clear in the Further Research section. We are happy to see that after the discussion period, the reviewers are in favor of our work, with both R1 and R4 raising the scores.

We thank all reviewers again for the high-quality comments and suggestions, which helped us tremendously in further improving the quality of our paper.

Best, ICLR 2022 Conference Paper3936 Authors

---

### Decision · Program_Chairs · 2022-01-20

**Decision:**

Reject

**Comment:**

The author response addressed some reviewer concerns, and generally reviewers increased their scores. However, there are important, and unanswered concerns about the generalization of the model. The discussion raised the concerns that despite the paper claim of "a specific class of higher order reasoning" emerging, the result suggests relatively simple strategies. This might not be a limitation of the approach, but of the evaluation scenario. So, this either requires a more nuanced view of the findings, and further empirical evidence to support the claim.